# Mice with Targeted Knockout of Tetraspanin 3 Exhibit Reduced Trabecular Bone Mass Caused by Decreased Osteoblast Functions

**DOI:** 10.3390/cells11060977

**Published:** 2022-03-12

**Authors:** Weirong Xing, Sheila Pourteymoor, Chandrasekhar Kesavan, Gustavo A. Gomez, Subburaman Mohan

**Affiliations:** 1Musculoskeletal Disease Center, VA Loma Linda Healthcare Systems, Loma Linda, CA 92357, USA; weirong.xing@va.gov (W.X.); sheila.pourteymoor@va.gov (S.P.); chandrasekhar.kesavan@va.gov (C.K.); gustavo.gomez2@va.gov (G.A.G.); 2Departments of Medicine, Loma Linda University, Loma Linda, CA 92354, USA; 3Departments of Biochemistry, Loma Linda University, Loma Linda, CA 92354, USA; 4Departments of Orthopedic Surgery, Loma Linda University, Loma Linda, CA 92354, USA

**Keywords:** Tspan3, knockout, bone mass, osteoblast, bone formation, bone

## Abstract

Tetraspanin3 (TSPAN3) was identified as a binding partner of claudin11 (CLDN11) in osteoblasts and other cell types. Mice with targeted disruption of *Cldn11* exhibited trabecular bone mass deficit caused by reduced bone formation and osteoblast function. To determine if the disruption of CLDN11 interacting protein gene *Tspan3* results in a similar skeletal phenotype as that of *Cldn11* knockout (KO) mice, we generated homozygous *Tspan3* KO and heterozygous control mice and characterized their skeletal phenotypes at 13 weeks of age. Micro-CT measurements of the secondary spongiosa of the distal femur revealed 17% and 29% reduction in trabecular bone volume adjusted for tissue volume (BV/TV) in the male and female mice, respectively. Similarly, trabecular BV/TV of the proximal tibia was reduced by 19% and 20% in the male and female mice, respectively. The reduced trabecular bone mass was caused primarily by reduced trabecular thickness and number, and increased trabecular spacing. Consistent with the reduced bone formation as confirmed by histomorphometry analyses, serum alkaline phosphatase was reduced by 11% in the KO mice as compared with controls. Our findings indicate that TSPAN3 is an important positive regulator of osteoblast function and trabecular bone mass, and the interaction of TSPAN3 with CLDN11 could contribute in part to the bone forming effects of Cldn11 in mice.

## 1. Introduction

Tetraspanins are a family of membrane proteins that have four transmembrane alpha-helices and two extracellular domains, one short and one longer, with typically 100 amino acid residues. The transmembrane superfamily proteins are considered to act as scaffolding proteins, anchoring multiple proteins consisting of adhesion, signaling, and adaptor proteins to one area of the cell membrane [1]. It is believed that tetraspanin (TSPAN) regulates adhesion-mediated (integrins/FAK), receptor-mediated (EGFR, TNF-α, c-Met, c-Kit), and intracellular signaling (PKC, PI4K, β-catenin) [2]. Of this superfamily, TSPAN3 protein has been shown to interact with claudin11 (CLDN11) in oligodendrocytes, where it was originally identified as a CLDN11-associated protein in a yeast two-hybrid screen using CLDN11 as bait [3]. In our previous studies, we found that *claudin (Cldn)11* expression was increased by more than 50-fold during ascorbic acid-induced differentiation of mouse calvarial osteoblasts [4]. Mice with disruption of *Cldn11* exhibited a low bone mass phenotype. Trabecular bone mass of the femur of the *Cldn11* knockout (KO) mice was reduced by 40%, which was primarily caused by reduced osteoblast differentiation and impaired bone formation [4]. To investigate the mechanism for Cldn11 effects on osteoblast differentiation, we executed reciprocal immunoprecipitation assays with antibodies against CLDN11 and TSPAN3, respectively. We found that CLDN11 was associated with TSPAN3 in osteoblasts. Immunohistochemistry analyses confirmed that CLDN11 and TSPAN3 were expressed similarly in the lining cells of the trabecular bone of the femur [4]. In vitro studies showed that *Tspan3* expression was also increased during osteoblast differentiation with a similar biphasic expression pattern to that of *Cldn11*. Furthermore, we found that knockdown of *Tspan3* expression by lentivirus-mediated shRNA resulted in reduced expression of markers of osteoblast differentiation [4]. Our studies indicated that TSPAN3 was involved in the regulation of osteoblast differentiation and function in vitro. However, the role it plays in osteoblast differentiation and bone formation in vivo remains unknown.

Based on our previous in vitro findings that TSPAN3 functions similarly to CLDN11 in regulating osteoblast differentiation, we predicted that CLDN11–TSPAN3 interaction would be important in mediating CLDN11 biological effects in osteoblasts. To test if CLDN11–TSPAN3 interaction is critical in mediating CLDN11 actions in bone, we generated *Tspan3* KO mice and determined if mice with the disruption of *Tspan3* exhibits trabecular bone mass deficit like that of *Cldn11* KO mice.

## 2. Materials and Methods

### 2.1. Tspan3 KO Mice

*Tspan3* KO mice were generated as described [5]. Breeding pairs of *Tspan3* KO mice were kindly provided by Dr. Tannishtha Reya at the University of California San Diego School of Medicine and bred with C57BL/6 mice for several generations at the animal facility of VA Loma Linda Healthcare Systems. Homozygous *Tspan3* KO male mice were bred with heterozygous females to obtain homozygous KO and littermate heterozygous controls for skeletal phenotype evaluation. Mice were genotyped using PCR and euthanized at 13 weeks of age for skeletal phenotype evaluation. Heterozygous controls were used as controls to enable us to use a breeding scheme (heterozygous x homozygous mutant) which would allow for the generation of a higher number of homozygous mutant mice. Animals were housed at the Veterans Administration Loma Linda Healthcare System (VALLHS) according to approved standards with controlled temperature (22 °C) and illumination (14 h light, 10 h dark), as well as unlimited food and water. Animal procedures were performed by a protocol (MOH0007/00014) approved by the Institutional Animal Care and Use Committee of the VALLHS. Mice were anesthetized with anesthetics (isoflurane) prior to the procedures. The animals were euthanized by exposure to carbon dioxide followed by cervical dislocation.

### 2.2. Micro-CT Evaluation

Femurs and tibias isolated from 13-week-old mice were scanned by X-ray at 55 kVp volts for trabecular bone at a resolution of 10.5 µm/slice. Trabecular bone parameters were measured at the secondary spongiosa region of distal femur and proximal tibia using micro-computed tomography (microCT, Scanco vivaCT40, SCANCO Medical AG, Zurich, Switzerland). For femurs, the trabecular region started at 0.36 mm from the distal growth plate in the direction of the metaphysis and extended for 2.25 mm. For tibias, the trabecular region started at 0.26 mm from the proximal growth plate and extended for 0.52 mm. The exact numbers and location of slices used for analyses were adjusted for length so that the analyzed regions were anatomically comparable between samples. Bone volume (BV, mm^3^), bone volume fraction (BV/TV, %), trabecular number (Tb. N, mm^−1^), trabecular thickness (Tb. Th, mm) and trabecular space (Tb. Sp, mm) were evaluated as reported [6,7,8]. Cortical bones at mid-diaphysis of the femur and the tibia were scanned at 70 kV energy and 114 µA intensity. Data were quantified from 200 slices (2.1 mm) of cortical bone.

### 2.3. Histology

Lone bones from 13-week-old mice were fixed in 10% formalin overnight, washed, dehydrated, and embedded in methyl methacrylate resin for sectioning. Tibia sections from males were stained with trichrome blue and safranin orange for histomorphometry analyses as reported [9]. Trabecular bone parameters of the secondary spongiosa were measured in a blinded fashion with computer software OsteoMeasure V3.1.0.2 (OsteoMetrics, Decatur, GA, USA) as described [10,11]. Adipocytes were counted in the trichrome blue-stained sections in a blinded fashion.

### 2.4. Serum Alkaline Phosphatase (ALP) Assay

Mouse serum level of ALP was measured as described previously [12].

### 2.5. Statistical Analysis

Data are presented as mean ± SEM from 7–9 replicates per group. Data were analyzed by Student’s *t*-test or two-way ANOVA as appropriate.

## 3. Results

### 3.1. Mice with Disruption of Tspan3 Exhibit Low Bone Mineral Density

Body weight and body length were not changed in 13-week-old *Tspan3* KO mice compared to gender-matched control mice (data not shown). Femur length was not significantly different in either male or female *Tspan3* KO mice at 13 weeks of age compared to corresponding gender-matched control heterozygous littermates (Figure 1A). Micro-CT analyses found that trabecular bone volume adjusted for tissue volume (BV/TV) was reduced by 29% and 17% in the femur of *Tspan3* KO female and male mice, respectively (Figure 1B,C). The reduced trabecular bone volume of the femur was due to the 14% and 9% reduction in trabecular number and 10% and 11% decrease in thickness, resulting in a 15% and 9% increase in trabecular separation in female and male KO mice, respectively (Figure 1D–F). The trabecular volumetric bone mineral density (BMD) was decreased by 21% and 12% in the femur of *Tspan3* female and male KO mice, respectively, compared to corresponding control mice (Figure 1G). The trabecular parameters exhibited similar changes in the tibia of *Tspan3* KO mice as those of the femur. The trabecular BV/TV of the tibia was 20% less in both *Tspan3* KO female and male mice compared to control mice (Figure 2A). In agreement with the bone mass of the femur, the decrease in trabecular bone mass of the tibia of *Tspan3* KO female and male mice was caused by 12% and 8% decreases in trabecular number and thickness, respectively, and a 10%–15% increase in trabecular separation (Figure 2B–D). Total volumetric BMD is reduced by 12% and 14% in the tibia of *Tspan3* KO female and male mice, respectively (Figure 2E). Analysis of data for gender differences revealed no significant gender difference in the trabecular bone parameters of *Tspan3* KO mice.

To identify the cause of reduced bone mass of *Tspan3* KO mice, we performed histology analyses of the proximal tibias derived from 13-week-old male mice. Bone sections were stained with trichrome and the histomorphometry of the trabecular bone of the secondary spongiosa was analyzed. As shown in Figure 3, osteoid area (O. Ar), osteoid perimeter (O. Pm), and osteoid width (O. Wi) were reduced by 47%, 30% and 27%, respectively, in the proximal tibia of the *Tspan3* KO males as compared with littermate control males (Figure 3A–D). While the trabecular parameter of BV/TV and trabecular thickness were not significantly changed, trabecular number was decreased when spacing was increased by 20% and trabecular spacing was increased by 36% in the *Tspan3* KO males (Figure 3E–H). There was no significant change in bone marrow adipocyte number in the *Tspan3* KO mice (Figure 3I). Consistent with the decreased bone formation evidenced by histology and micro-CT analyses, serum alkaline phosphatase (ALP) was reduced by 11% in the *Tspan3* KO mice compared to the control mice (Figure 3J).

### 3.2. Disruption of Tspan3 in Mice Does Not Influence Cortical Bone Mass

In previous studies, we found that the cortical bone phenotype was unchanged at the femur mid-diaphysis of *Cldn11* KO female mice at 12 weeks of age. To determine if the loss of *Tspan3* influences the cortical bone phenotype, cortical bone at mid-diaphysis of the femurs and tibias was scanned. Consistent with the *Cldn11* KO data, cortical bone parameters at femur mid-diaphysis were not significantly different between *Tspan3* KO female and control mice (Figure 4A–C). There was no change in the cortical BV/TV in the tibias of *Tspan3* KO mice (Figure 4D,E). However, volumetric BMD was significantly less in the *Tspan3* female KO mice at tibia mid-diaphysis compared to corresponding littermate control mice (Figure 4F). In the male mice, there were no significant differences in total volumetric BMD between the two genotypes at tibia mid-diaphysis. To determine if KO of *Tspan3* influences *Cldn11* expression, we measured mRNA levels of *Cldn11* in the bones of *Tspan3* KO mice and corresponding control mice. We found that there was no significant difference in *Cldn11* mRNA levels between the *Tspan3* KO and the control micemice (data not shown).

## 4. Discussion

TSPAN3 is known to mediate signal transduction events that regulate the expression of the A disintegrin and metalloproteinase 10 (ADAM10), presenilin and the amyloid precursor protein and the development and progression of acute myelogenous leukemia (AML) [5,13]. There are no published data linking TSPAN3 to bone density in humans. Mice with disruption of *Tspan3* displayed impaired leukemia stem cell self-renewal and disease propagation and markedly improved survival in mouse models of AML by a mechanism through which the response to CXCL12/SDF-f signal pathway is disabled [5]. The role of TSPAN3 in regulating bone homeostasis is unknown, although other family members have been implicated to play a role in mediating cytoskeletal rearrangement and sealing zone formation of osteoclast and osteoblast function [14,15]. Knockdown of *Tspan7* impaired osteoclast bone resorption function and Src and Pyk2 phosphorylation in osteoclasts [14]. The global deletion of *Tspan CD82* resulted in attenuated bone growth and enhanced bone marrow adipogenesis [15]. There is another report that TSPAN 12, 14 and 21 were involved in the promotion of bone morphogenetic protein signaling in *C. elegans* [16]. In this study, we characterized the skeletal phenotypes of *Tspan3* KO mice at 13 weeks of age. Our micro-CT measurements of the secondary spongiosa of distal femur revealed 17% and 29% reduction in trabecular BV/TV in the male and female mice, respectively. Similarly, trabecular BV/TV of the proximal tibia was reduced by 19% and 20% in the male and female mice, respectively. The reduced trabecular bone mass was caused primarily by reduced trabecular thickness and number, and increased trabecular separation. Serum ALP activity was reduced by 11% in the KO mice. Our histology measurements indicated that decreases in osteoid area, osteoid width and osteoid perimeter were consistent with a decrease in bone formation. Our data demonstrate that loss of *Tspan3* primarily influences the trabecular but not the cortical bone phenotype, as in the case of *Cldn11* KO mice [4].

While our findings suggest that TSPAN3 is an important positive regulator of osteoblast functions and trabecular bone mass and could contribute in part to the bone forming effects of CLDN11 in mice, the skeletal phenotypes are not as severe as those of *Cldn11* KO mice. The reduced trabecular bone mass deficit in *Tspan3* KO mice compared to *Cldn11* KO mice could be explained by a possible compensation by other members of TSPAN family proteins. It is known that the expression of *Tspan 5* and *7* could compensate for the functional loss of TSPAN3 in neuroblastoma cells and in the brains of TSPAN3-decient mice [13]. Further studies are needed to determine whether the CLDN11 also interacts with the TSPAN7 and to evaluate the relative contribution of TSPAN7–CLDN11 interaction in mediating the bone forming effects of CLDN11. The mechanisms by which TSPAN3 precisely regulates downstream of CLDN11 signal transduction in osteoblasts remains unclear. In other cell types, several TSPAN family members have been shown to promote Notch signaling by promoting physical association between γ-secretase and Notch in the membrane microdomain [13]. Future studies focused on how domains and motifs of TSPAN3 promote cellular signaling and whether CLDN11–TSPAN3 interaction promotes key signaling pathways such as BMP and Notch that are involved in osteoblast differentiation are needed to understand the specific mechanisms of TSPAN3 function in osteoblasts. In addition, it is necessary to use sophisticated imaging techniques to examine the spatiotemporal interactions mediated by TSPAN3/7 and CLDN11 and define how the scaffolding properties of TSPAN3/7 proteins contribute to the formation and stabilization of Notch signal transduction complexes at the plasma membrane of osteoblasts and osteoclasts in bone.

## Figures and Tables

**Figure 1 cells-11-00977-f001:**
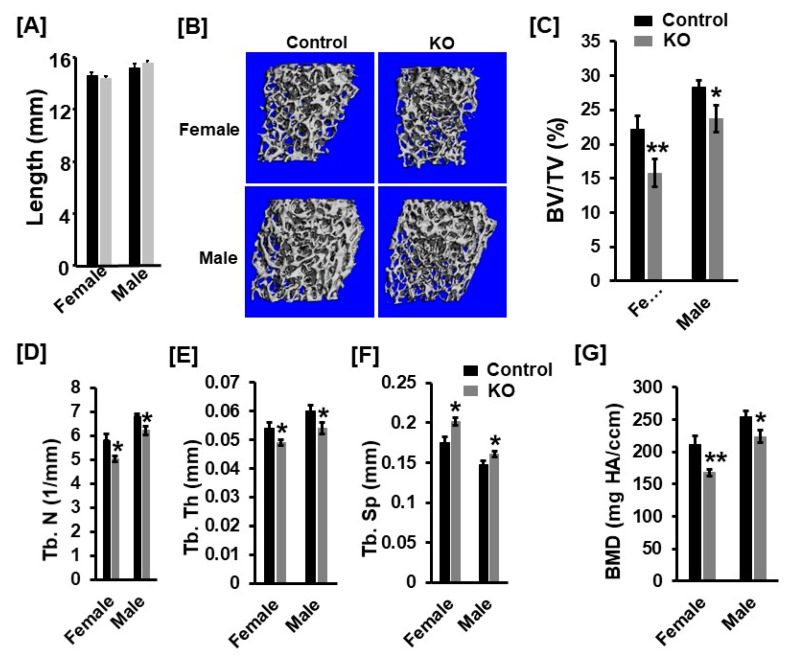
Mice with disruption of *Tspan3* exhibit low bone mineral density in the distal femurs. (**A**) Femur length of the *Tspan3* knockout (KO) and the control littermates at 13 weeks of age. (**B**) Micro-CT images of the trabecular bone of the distal femurs of 13-week-old *Tspan3* KO and control littermates (**C**–**G**) Quantitative micro-CT data of the trabecular bone of the distal femur. TV, tissue volume; BV, bone volume; Tb. N, trabecular number; Tb. Th, trabecular thickness; Tb. Sp, trabecular spacing; vBMD, volumetric bone mineral density. Values are mean ± SEM (*n* = 7–9). *: *p* < 0.05; **: *p* < 0.01 as compared to the control littermates.

**Figure 2 cells-11-00977-f002:**
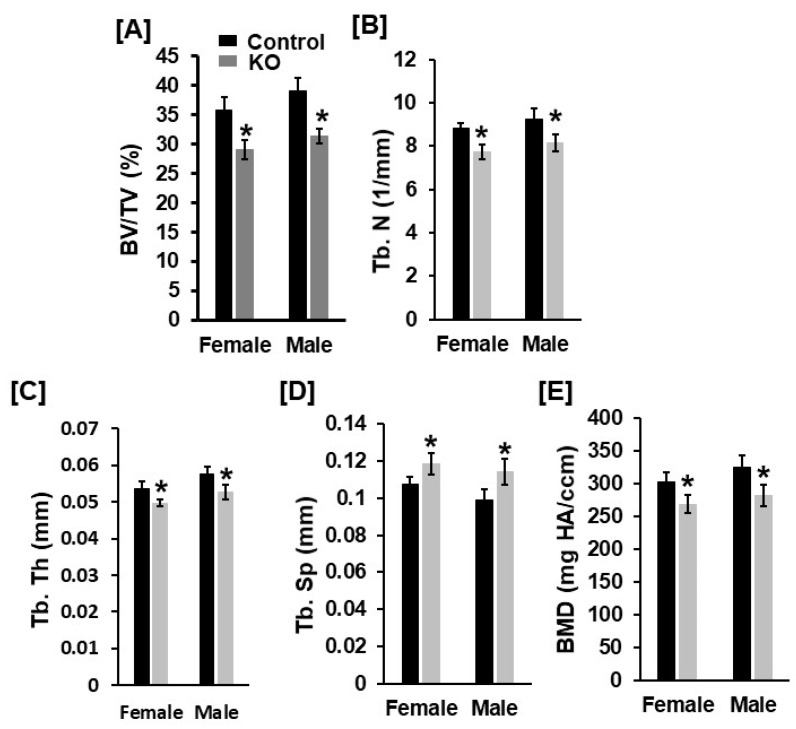
Mice with disruption of *Tspan3* exhibit low bone mineral density in the proximal tibias. (**A**–**E**) Quantitative micro-CT data of the trabecular bone of the proximal tibia. TV, tissue volume; BV, bone volume; Tb. N, trabecular number; Tb. Th, trabecular thickness; Tb. Sp, trabecular spacing; vBMD, volumetric bone mineral density. Values are mean ± SEM (*n* = 7–9). *: *p* < 0.05 as compared to the control littermates.

**Figure 3 cells-11-00977-f003:**
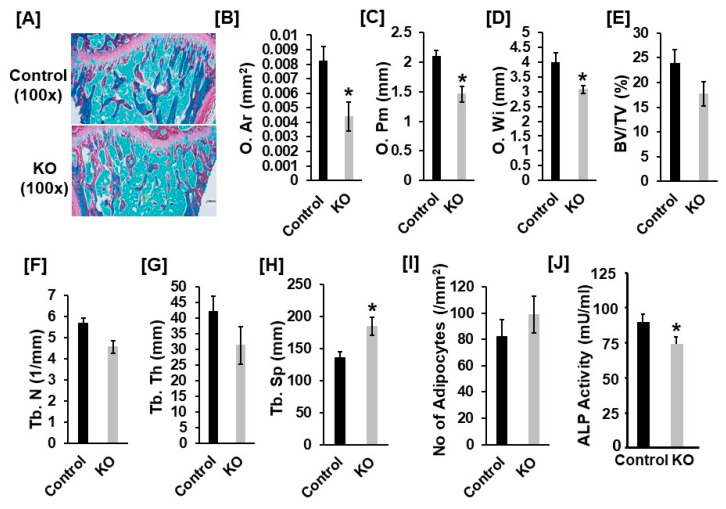
Reduced bone formation in *Tspan3* KO male mice. Bone sections were stained with trichrome and the histomorphometry of the trabecular bone of the secondary spongiosa of the tibia was analyzed. (**A**) Representative images of longitudinal sections of the proximal tibial metaphysis, stained with trichome blue and Safranin Orange. (**B**–**H**) Quantitative histomorphometry data of the trabecular bone of the proximal tibial metaphysis. O. Ar, osteoid area; O. Pm, osteoid perimeter; O. Wi, osteoid width; TV, tissue volume; BV, bone volume; Tb. N, trabecular number; Tb. Th, trabecular thickness; Tb. Sp, trabecular spacing; Values are Mean ± SEM (*n* = 7–9). (**I**) Number of adipocytes in tibial bone marrow section. (**J**) Serum alkaline phosphatase (ALP) activity. *: *p* < 0.05 as compared to the control littermates.

**Figure 4 cells-11-00977-f004:**
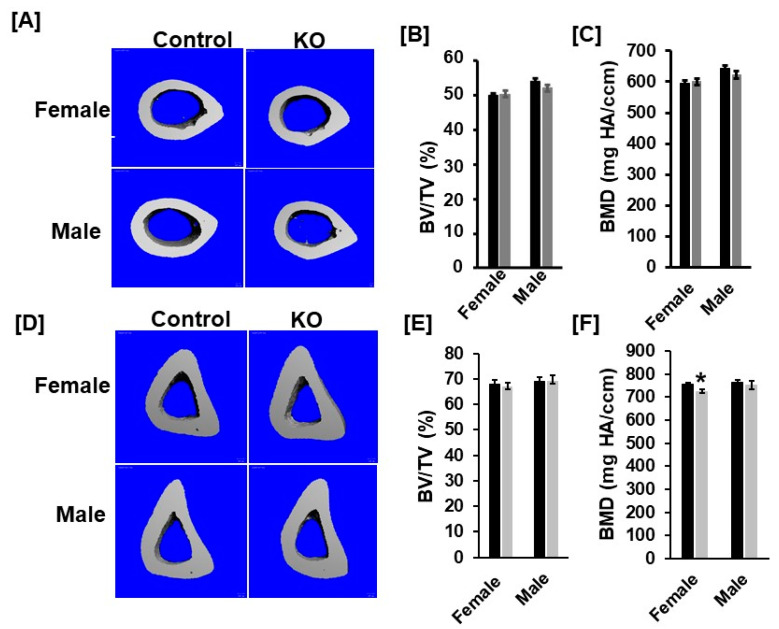
Cortical bone parameters at the femur and tibia mid-diaphysis are not significantly different between *Tspan3* KO and control mice. (**A**) Micro-CT images of the cortical mid-diaphysis of the femurs of 13-week-old *Tspan3* KO and control littermates. (**B**,**C**) Quantitative micro-CT data of the femur cortical bone parameters of the *Tspan3* KO and their control littermate mice. (**D**) Micro-CT images of the cortical mid-diaphysis of the tibias of 13 weeks old *Tspan3* KO and control littermates. (**E**,**F**) Quantitative micro-CT data of the tibia cortical bone parameters of the *Tspan3* KO and their control littermate mice. TV, tissue volume; BV, bone volume; vBMD, volumetric BMD. Values are Mean ± SEM (*n* = 6–8). *: *p* < 0.05 as compared to the control littermates.

## Data Availability

The raw datasets generated and/or analyzed during the current study are available from the corresponding author on reasonable request.

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
