# Peer review of "Mice with Targeted Knockout of Tetraspanin 3 Exhibit Reduced Trabecular Bone Mass Caused by Decreased Osteoblast Functions"

_cells, 2022, doi:10.3390/cells11060977_

Round 1
Reviewer 1 Report
The authors have sufficiently substituted the additional information. As a result, the work is improved, and the flow is also good. Additionally, there are no concerns since the authors have integrated and offered sufficient responses to the questions that were expressed. Therefore, following a thorough examination of the whole manuscript, I am satisfied with the current revision.
Reviewer 2 Report
The authors have addressed my points.
There are still a few typos, for example
See sentence line 287
See sentence line 291 - “Fig I” should be “Fig 3I”
See sentence line 362
See sentence line 356 – “CXCL12/SDF-f”
See sentence line 444 – “Tsapn3/7”
This manuscript is a resubmission of an earlier submission. The following is a list of the peer review reports and author responses from that submission.
Round 1
Reviewer 1 Report
In the present work, Xing et al showed that the targeted knockout of Tspan3 resulted in the reduced trabecular bone, simultaneously in the reduction of the osteoblast function. Further, the authors concluded that Tspan3 is an important positive regulator of osteoblast functions and trabecular bone mass and could contribute in part to the bone-forming effects of Cldn11 in mice.
Line 38-39; I am curious did the authors of reference 3 identify only this interaction and no other with Cldn-11.
Line 43-46; How did the authors choose to select the Cldn-11 and Tspan3 for immunoprecipitation assays? What is the reason that other receptors protein family members could not be supposed to work with Cldn-11?
Line 97: change the font format and keep it consistent throughout the manuscript.
Based on the literature and previous study observation, the authors performed this experiment. The connection with bone mass changes is interesting but at the same time, it is very important to report that the KO of Tspan3 has what kind of effect on Cldn-11. I would strongly recommend that authors should perform the immunoprecipitation in KO mice and show at least western blot to detail the connection before describing the phenotypic effect.
On the same note, authors should report the localization of both the proteins in Tspan3 KO mice. I would strongly recommend including these experiments and discussing the finding.
Results should include the strong finding to correlate Cldn-11 and Tspan3 connection with different experiments. I understand that the manuscript is communication but not the complete research article but based on the information provided I suggest authors provide these data.
Minor comments
The abstract description of the previous study is not very useful. It is useful to keep the information in the introduction section as already provided. Also, the information about the knockdown in vitro is confusing. I recommend rewriting the abstract with the flow of statements such as rationale for study with the important findings.
Reviewer 2 Report
The authors investigate the effect of Tspan3 knock-out on the function of osteoblasts and its effects on bone in mice.
Strength: the authors use imaging techniques to show that Tspan 3 KO has a definitive effect on long bones.
Weakness: a molecular mechanism is not provided for the observed Tspan 3 KO phenotype.
- The authors state “Knockdown of Tspan3 expression by lentivirus mediated shRNA resulted in reduced expression of markers of osteoblast differentiation in vitro.” – this data should be included in the manuscript, or a reference to the work provided.
- In Figure 3, can you please show representative histology sections.
- Did you observe any change in bone marrow adipogenesis with Tspan 3 KO in vivo, or in vitro (with shRNA)?
- Did you observe any change in osteoclast activity in the Tspan 3 KO mice vs wild type, in vivo.
- Did you observe any change in BMP signaling or can you report on any other molecular pathway affected by Tspan 3?
- Did you observe any other obvious phenotypic changes between Tspan 3 KO mice and wild type animals? Is there a difference in weight?
- Are there any mutations in Tspan 3 that occur in humans that may be relevant to your observations in mice? Any related to skeletal defects?
Some typos:
-Introduction, line35 “adhesion-medicated” should be “adhesion-mediated”.
-line 146 has a space missing between “the” and “Tspan 3”.
-In Discussion, line 184 “progpagation” should be “propagation”, Tspan “DC82”, should be “CD82”.
-Some of the Figure captions have extra period at the end.